# Antagonism between viral infection and innate immunity at the single-cell level

**Frederic Grabowski**[1]☯, **Marek Kochańczyk**[1]☯, **Zbigniew Korwek**[1], **Maciej Czerkies**[1], **Wiktor Prus**[1], **Tomasz Lipniacki**[1,2]*

1 Institute of Fundamental Technological Research, Polish Academy of Sciences, Warsaw, Poland,
2 Department of Statistics, Rice University, Houston, Texas, United States of America

☯ These authors contributed equally to this work.
* tlipnia@ippt.pan.pl

**Data Availability Statement:** Source code of the agent-based simulator of the innate immune response to an infection with an RNA virus is available at https://github.com/grfrederic/visavis

## Abstract

When infected with a virus, cells may secrete interferons (IFNs) that prompt nearby cells to prepare for upcoming infection. Reciprocally, viral proteins often interfere with IFN synthesis and IFN-induced signaling. We modeled the crosstalk between the propagating virus and the innate immune response using an agent-based stochastic approach. By analyzing immunofluorescence microscopy images we observed that the mutual antagonism between the respiratory syncytial virus (RSV) and infected A549 cells leads to dichotomous responses at the single-cell level and complex spatial patterns of cell signaling states. Our analysis indicates that RSV blocks innate responses at three levels: by inhibition of IRF3 activation, inhibition of IFN synthesis, and inhibition of STAT1/2 activation. In turn, proteins coded by IFN-stimulated (STAT1/2-activated) genes inhibit the synthesis of viral RNA and viral proteins. The striking consequence of these inhibitions is a lack of coincidence of viral proteins and IFN expression within single cells. The model enables investigation of the impact of immunostimulatory defective viral particles and signaling network perturbations that could potentially facilitate containment or clearance of the viral infection.

## Author summary

When our cells fight a proliferating virus, the virus fights back. We created a spatial stochastic computational model to understand the impact of such mutually antagonistic relations on the progression of viral infection. The model was tuned to an array of in vitro experiments with the respiratory syncytial virus infecting a monolayer of cells originating from the human respiratory tract. It reproduces complex spatial patterns of cell signaling states. We observed that the antagonistic virus–host cell interactions elicit dichotomous cell fates: infected cells either express viral proteins or produce interferons (biochemical messengers that alert not yet infected cells).

## Introduction

Innate immunity provides the first line of defense against viral infections. It acts primarily by feed-forward signaling from infected cells to not (yet) infected bystander cells: upon

(under the BSD-3-Clause license). The simulator implements the computational model described in and used throughout the article. The immunostaining imaging dataset is available at Zenodo (doi: 10.5281/zenodo.7428925). The ELISA, dPCR and Western Blot data are available as supplementary files.

**Funding:** This study was funded by Narodowe Centrum Nauki (National Science Centre, Poland) grant 2018/29/B/NZ2/00668 (TL) and the Norwegian Financial Mechanism GRIEG-1 grant operated by Narodowe Centrum Nauki (National Science Centre, Poland) 2019/34/H/NZ6/00699 (TL). The funders had no role in study design, data collection and analysis, decision to publish, or preparation of the manuscript.

**Competing interests:** The authors have declared that no competing interests exist.

recognition of viral genetic material by intracellular receptors, some infected cells synthesize and secrete intercellular messengers, most importantly interferons (IFNs), that prompt nearby cells to enter an antiviral state, rendering them more resistant to secondary infections [1–5]. Imminently, with respect to the presence or absence of the virus and the antiviral state, a (partially) infected cell population is spatiotemporally stratified into functionally distinct subpopulations [3,6–8].

To invade a host cell and successfully replicate, viruses evolved multiple strategies to evade or counteract cellular mechanisms of the innate immune response [9]. Commonly, viral, often non-structural, proteins attenuate viral RNA sensing and interfere with the synthesis of IFNs and IFN-induced STAT signaling. For example, in respiratory viruses, NS1 of influenza A/B virus (IAV/IBV) directly and indirectly inhibits activation of IRF3, a key transcription factor of type I and type III IFNs [10,11]; NS1 and NS2 of the respiratory syncytial virus (RSV) similarly impinge on activation of IRF3 and, principally by disrupting STAT1 and STAT2 signaling, impede the induction of IFN-stimulated genes (ISGs) [12–14]; a multitude of effectively analogous interactions have been recently discovered in the case of SARS-CoV-2 [15,16]. The mutual virus–host cell interactions amplify heterogeneity of the stratified, infected cell population and modulate the progression of viral infection. The picture can be even more convoluted in the presence of immunostimulatory, replication-incompetent defective viral particles [17].

Understanding the complex interplay between viruses and the innate immune response at the system level requires experimentation at the single-cell resolution combined with data-driven single-cell level computational modeling [18,19]. Thus far, integration of quantitative data on *Ifnb1* (IFNβ gene) and ISGs expression in single cells over time and stochastic modeling allowed to demonstrate that paracrine signaling has a major impact on heterogeneous cell responses to infection [2,3,20]. Recently, benefits of interferon expression heterogeneity have been studied within a spatial agent-based model of an infection with a "generic" virus [21]. A spatial agent-based model with ODE-based intracellular kinetics and stochastic transitions between cell types in a recent work of Aponte-Serrano *et al.* [22] was directly tuned to recapitulate the kinetics of plaque formation by IAV in human bronchial cells. It was demonstrated that a sufficiently fast synthesis and diffusion of (or a prestimulation with) interferon may entirely arrest plaque growth. The action of IAV NS1 was factored into a reduction of the activity of a viral RNA sensor and an IRF inducer, RIG-I. The employed generative computational approach enabled extending the model by inclusion of multiple types of immune cell types (also those involved in adaptive immunity), several diffusible cytokines, as well as cell attachment and migration [23,24]. It was proposed that locally concentrated exposure to a virus can elicit a productive infection, whereas uniform exposure to the virus is likely ineffective.

Here, we present an agent-based, spatial, stochastic model of infection propagation in a monolayer of cells. The model was developed and manually calibrated based on an array of our experiments on epithelial cells of respiratory origin (A549 cell line) infected with human RSV, a common pathogen in severe respiratory disease in both young children and the elderly [25–27]. Upon viral infection, these cells communicate through interferons of type I (IFNβ) and type III (IFNλs) [28]. By analyzing results from cell population-level techniques (Western blot, ELISA, dPCR), we delineated the structure and constrained kinetic rates of the regulatory network. By analyzing heterogeneous single-cell responses in immunofluorescence microscopy images, we quantified the degree of antagonism between the virus and host cells. The data demonstrate that, by triggering universal interferon-mediated mechanisms of the innate immune response [28,29], epithelial cells fight the virus using STAT-inducible ISGs, which inhibit the synthesis of viral RNA and proteins. RSV proteins, in turn, specifically attenuate the immune response by interfering with the activation of IRF3 [30,31], synthesis of IFNs [32],

phosphorylation of STAT1 [33], and by degrading STAT2 [34–36]. These reciprocal, antagonistic interactions give rise to a transient switch-like behavior: an infected cell is unlikely to simultaneously produce IFNs and express RSV proteins. The modes of interference of various viruses display common, recurrent motifs, rendering the constructed model general enough to study, after readjustment of transition rates, kinetics of infection with another virus.

## Results

### Spatial stochastic model of viral infection

The constructed agent-based, spatial, stochastic model of viral infection is outlined in Fig 1. Below, we describe modeled cell components and biological processes, including virus–host cell interactions. Model derivation and parameter calibration are described in Methods. Experimental results used to parametrize the rates of processes included in the model are shown in Figs A–E in S1 Appendix (we provide raw data as S1, S2, and S3 Dataset files). Model processes and rate parameters are given in Table A in S1 Appendix.

Agents (cells) are arranged in a regular two-dimensional lattice. The state of each agent-cell is fully characterized by seven discrete variables: the binary variable v, which indicates the cell's infection status, and six other variables: vRNA, vProteins, pIRF3, IFNi, pSTATs, and ISGs, each assuming a discrete value from the set {0, 1, 2, 3} and in this way describing the status of the respective intracellular biochemical species (Fig 1A). The cell state evolves as a

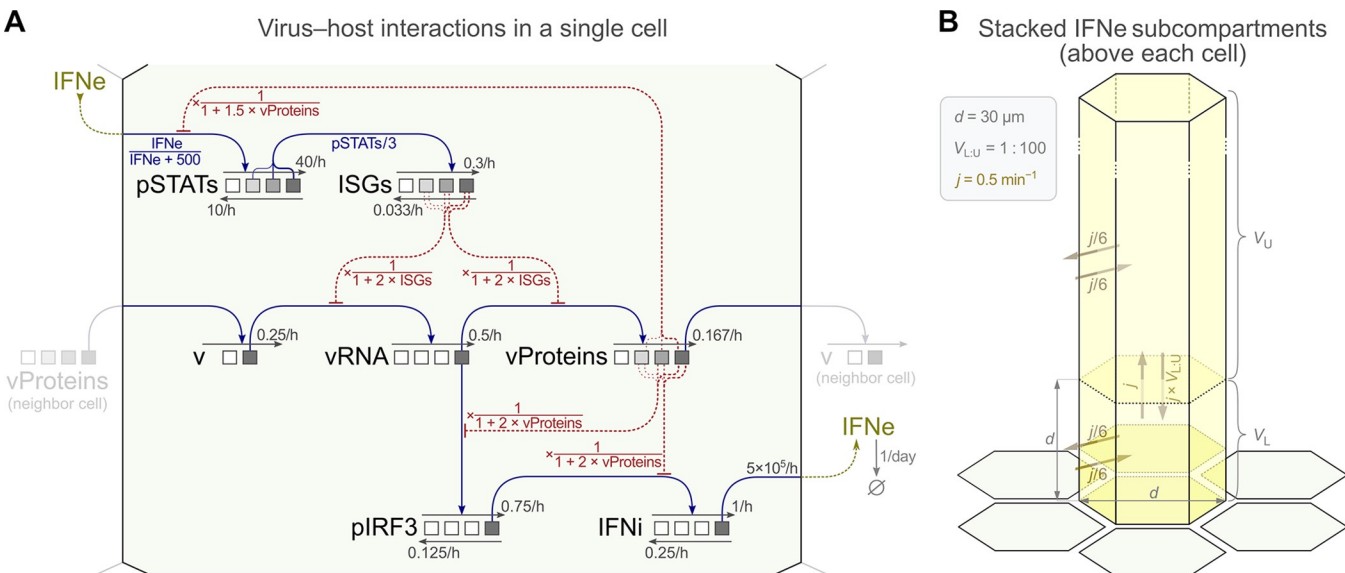

**Fig 1. Computational model. A.** Host cell–virus interactions and adjusted kinetic rate parameter values. Black arrows indicate state transitions of seven variables characterizing the cell state: v, vRNA, vProteins, pIRF3, IFNi, pSTATs, ISGs. All forward transitions (thin straight arrows pointing right) are induced by other variables (as indicated by blue arrows), while the reverse transitions (thin straight arrows pointing left) have constant rates. The corresponding rate coefficients are given in gray. For forward reactions to occur, the inducing agent variable has to be maximally active, with the exception of activation of pSTATs (Michaelis–Menten kinetics) and activation of ISGs (by partially active pSTATs); see model reactions in Table A in S1 Appendix. Red dashed hammer-headed arrows are inhibitory interactions: three of them emanate from vProteins, two emanate from ISGs. The expressions in red indicate how vProteins or ISGs modify the (forward) transition propensities. When an arrow has three roots (all inhibitions and ISGs activation), the strength of the interaction depends on the stage of activation of the inducing variable (additionally indicated by the hue of the inducing variable). The extracellular interferon IFNe is produced at the given rate (molecules / hour) when the state of the intracellular interferon is IFNi = 3. IFNe diffuses (see panel B) and, when present in the lower subcompartment above a given cell, regulates the pSTATs transition rate with Michaelis–Menten kinetics. An infected cell with vProteins = 3 infects adjacent (non-infected) cells (v = 0 → v = 1) at the given rate (the infecting virions are implicit). **B.** Processes related to the transport of extracellular IFN (IFNe) and related parameter values. IFNe diffuses between the lower and the upper subcompartment above each cell and between respective subcompartments above neighboring cells.

time-continuous Markov process influenced by neighboring cells, which may infect the cell ($v = 0 \rightarrow v = 1$), and by extracellular interferon, `IFNe`.

## Intracellular state transitions

As depicted in Fig 1A, viral infection ($v = 1$) leads to the synthesis of viral RNA (increase of `vRNA`), followed by the synthesis of viral proteins (increase of `vProteins`) and the production of infectious virions (implicit in the model). Advancements of viral entities (`v`, `vRNA`, `vProteins`) are irreversible. The emergence of viral RNA leads to phosphorylation of IRF3 (increase of `pIRF3` at `vRNA = 3`), which in turn leads to the synthesis of (intracellular) interferon (increase of `IFNi` at `pIRF3 = 3`) and its secretion. Secreted interferon is represented as a continuous variable (`IFNe`) and diffuses between subcompartments associated with individual cells (see Fig 1B and Methods for details). Extracellular interferon induces phosphorylation of STAT1 and STAT2 (represented by an increase of `pSTATs`, with saturable kinetics) that jointly trigger the synthesis of proteins coded by interferon-stimulated genes (increase of `ISGs`). In the model, we consider only one generic type of interferon that accounts for both type I and type III interferons (IFNβ and IFNλs, respectively; our experiments indicate that in the case of infection with RSV, IFNβ plays a decisive role, see Fig D panels a and b in S1 Appendix in conjunction with Fig A panels a, b and Fig E in S1 Appendix). It is worth noting that our model includes both autocrine and paracrine interferon signaling, however, as explained later, we have found that the autocrine interaction does not play a significant role during RSV infection of A549 cells.

**Virus–host cell interactions.** The model includes two mutually antagonistic groups of interactions. The first group comprises a three-level inhibition of the immune response—phosphorylation of IRF3, synthesis of interferons, and phosphorylation of STAT1 and STAT2—by viral proteins (the strengths of each inhibition increase with `vProteins`). The second group consists of two interactions, in which the accumulation of viral RNA and viral proteins is slowed down due to activity of proteins coded by interferon-stimulated genes (the strength of both inhibitions increases with `ISGs`). The presence of these two reciprocally antagonistic groups of interactions ensures the mutual inhibition of the virus and the innate immune response.

## Model simulations *vs*. immunostaining images

To characterize the incidence and co-incidence of biological entities included in the model at the single-cell level, we performed an array of experiments on A549 cell cultures infected with RSV (with or without additional IFNβ prestimulation). The cells were immunostained for RSV proteins, IRF3, phospho-Tyr701 STAT1 (p-STAT1), and IFNβ, auxiliarily stained with a DNA marker, and imaged using multi-channel confocal microscopy. In Fig 2, representative microscopy images from experiments at a multiplicity of infection (MOI) of 0.01 (Fig 2A and 2B) are juxtaposed with model simulations (Fig 2C). The case of an MOI of 1 is shown in Fig F in S1 Appendix.

Fig 2A and 2B show that at 8–10 h post infection (p.i.) with RSV, the infection may manifest through IRF3 phosphorylation (p-IRF3 is translocated to the nucleus, giving a discernible nuclear IRF3 signal, overlapping with that of NF-κB at later time points in Fig G in S1 Appendix), expression of RSV proteins, and/or production of IFNβ. In Fig 2A, a 2-hour-long treatment with brefeldin A prior to fixation enabled capturing IFNβ in the cytoplasm, allowing for its reliable visualization and quantification (see Fig H in S1 Appendix for immunostaining of IFNβ without a prior treatment brefeldin A). Initially, characteristic rings of p-STAT1 positive cells are formed (Fig 2A, 8 h p.i), often revealing an infected cell being a local source of IFNβ

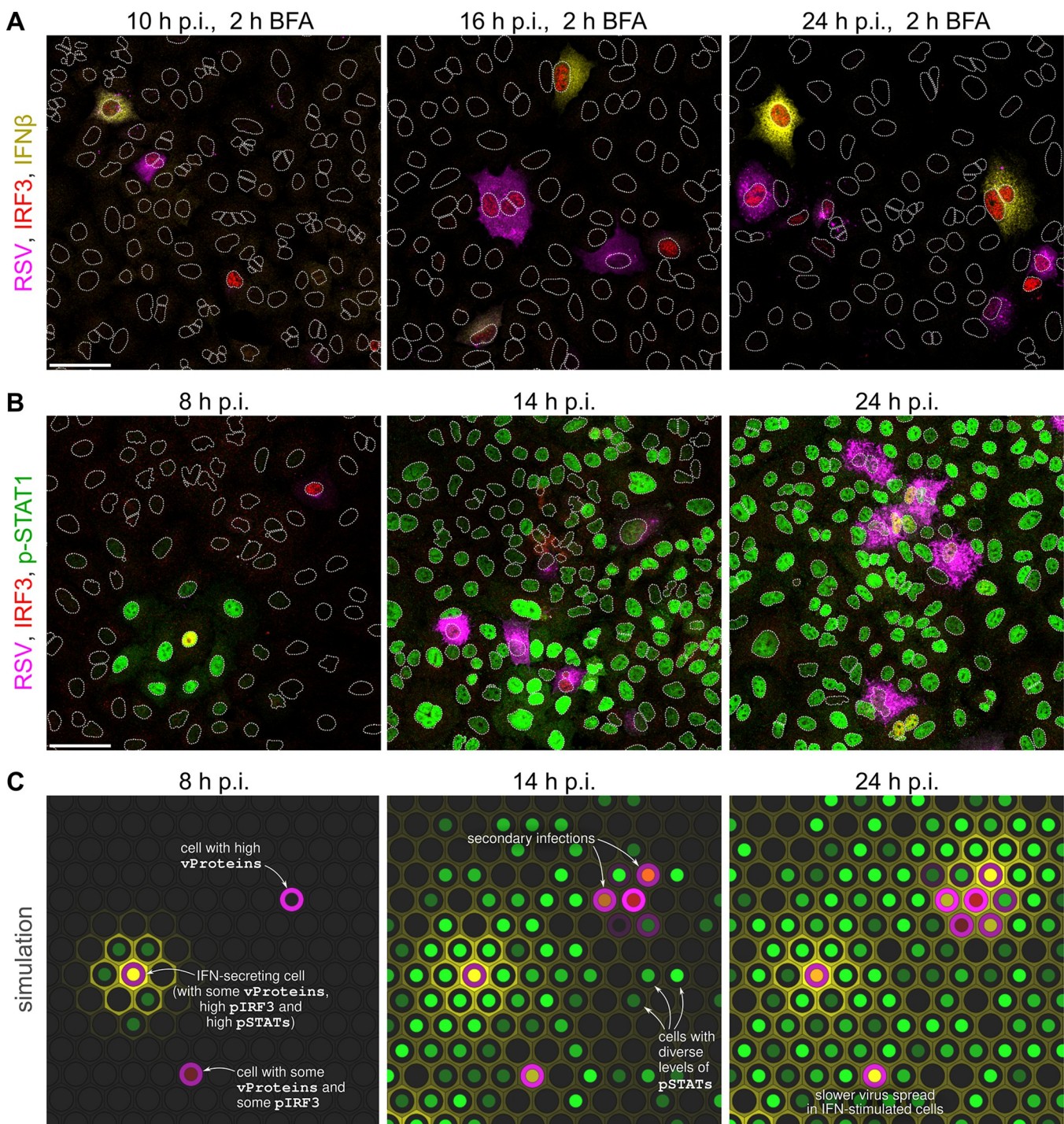

**Fig 2. Images from experiments and snapshots from a simulation. A, B.** Overlays of images of A549 cells at three time points post infection (p.i.) with RSV at an MOI of 0.01, immunostained for: RSV proteins (polyclonal antibody)–magenta, IRF3 –red, (intracellular) IFNβ –yellow (only in panel A), phospho-Tyr701 STAT1 (p-STAT1)–green (only in panel B). To improve the detection of intracellular IFNβ, we used a 2-hour-long treatment with brefeldin A (BFA, in panel A) before cell fixing, which prevented secretion of IFNβ produced during this treatment. When phosphorylated, IRF3 translocates to the nucleus giving a discernible signal. White dotted lines are nuclear outlines determined based on DAPI counterstaining (channel not shown). Scale bars, 50 μm. **C.** Snapshots from a simulation of infection at an MOI of 0.01 in a compact monolayer of cells (subpanels show fragments of a simulated 100 × 100 lattice). The color key corresponds to pseudocolors of immunostained proteins used in panels A, B (concentration of IFNβ in the lower subcompartment is indicated at hexagon borders; yellow/orange/olive colors in nuclei result from mixing of red and green at various intensities).

(Fig 2B). In subsequent time points, 14 h p.i. and 24 h p.i., secondary infections yield small clusters of RSV proteins-producing cells. In the considered time span, IFNβ reaches all the cells and the spatial profile of p-STAT1, still concentrated around local sources of IFNβ at 14 h p.i., is largely homogeneous across bystander cells at 24 h p.i. Fig 2C shows an example model simulation, at the same stages of infection. Contrary to immunostaining images, the simulation allowed us to study the same cell population at different time points, as well as to simultaneously record all components of the model.

After a high-MOI infection (Fig F in S1 Appendix), nearly all bystander cells display p-STAT1 already at 10 h p.i., but also very likely have a virus-infected neighbor. These cells are expected to be granted less time to upregulate their ISGs prior to infection compared to the bystander cells after low-MOI infection. As demonstrated using model trajectories (Fig I in S1 Appendix), during a low-MOI infection, the primary infected cells exhibit fast growth of the average level of vProteins and have low levels of ISGs (Fig I panel a in S1 Appendix), however the cells infected between 16 and 24 h p.i. (secondary infections) have on average a higher level of ISGs, which noticeably hamper the production of vProteins (Fig I panel b in S1 Appendix).

## Antagonism between RSV and the single-cell immune response

**RSV proteins terminate STAT activity.** Immunostaining images obtained at 24 h p.i. show markedly lower levels of p-STAT1 in cells expressing RSV proteins (Fig 3A). To quantify this effect, we compared the distributions of the nuclear p-STAT1 signal in cells stratified into either expressing or not expressing RSV proteins ('same cells' statistics in Fig 3B). Next, we selected cells that do not express RSV proteins, divided them into either having or not having an RSV proteins-expressing cell in direct neighborhood, and compared nuclear p-STAT1 between these two groups ('neighboring cells' statistics in Fig 3B). To characterize quantitatively the extent to which the considered distributions are discordant, we calculated the signed Kolmogorov–Smirnov (sKS) statistic (see Methods for details). The sKS statistic is a real number in the range $[−1, +1]$, where values near zero imply that the distributions are similar, low values suggest the first distribution frequently has lower values than the second distribution, and high values suggest that the second distribution frequently has higher values than the first one. We use the sKS statistic rather than a pure KS statistic because in certain cases the compared distributions exchange their locations. It also allows us to differentiate between an sKS value, which is consistently low and positive or low and negative for all experimental replicates, and a noisy sKS value changing its sign between replicates. We acknowledge that the sKS can be problematic when a narrower distribution is nested within a broader one, however, this is not the case in our study. For all studied MOIs (0.01, 0.1, and 1), the value of sKS is about 0.5 for 'same cells' and close to 0 for the 'neighboring cells', which agrees well with mean model predictions (Fig 3C). However, experimental results show high variability between replicates. To investigate the potential source of this variability, we first ran the model 10,000 times on a $50 \times 50$ cell lattice (containing $n = 2500$ cells, which is typical for shown experimental replicates) and computed the 95% CrI. The resulting plot shows that most of the variability in experiments with a low fraction of RSV proteins-expressing cells can be explained by stochastic noise. Next, we ran the model on a $50 \times 50$ lattice with parameters randomly selected from log-normal distributions (σ = 0.2) centered at the default values, and again showed the 95% CrI. The high variability of obtained numerical results demonstrates the systems' sensitivity to variation of underlying biological parameters, which could explain the spread observed in experimental replicates. Snijder *et al.* showed that cell population context is a major component of the cell-to-cell variation observed during viral infection [37]. Using the model, in Fig

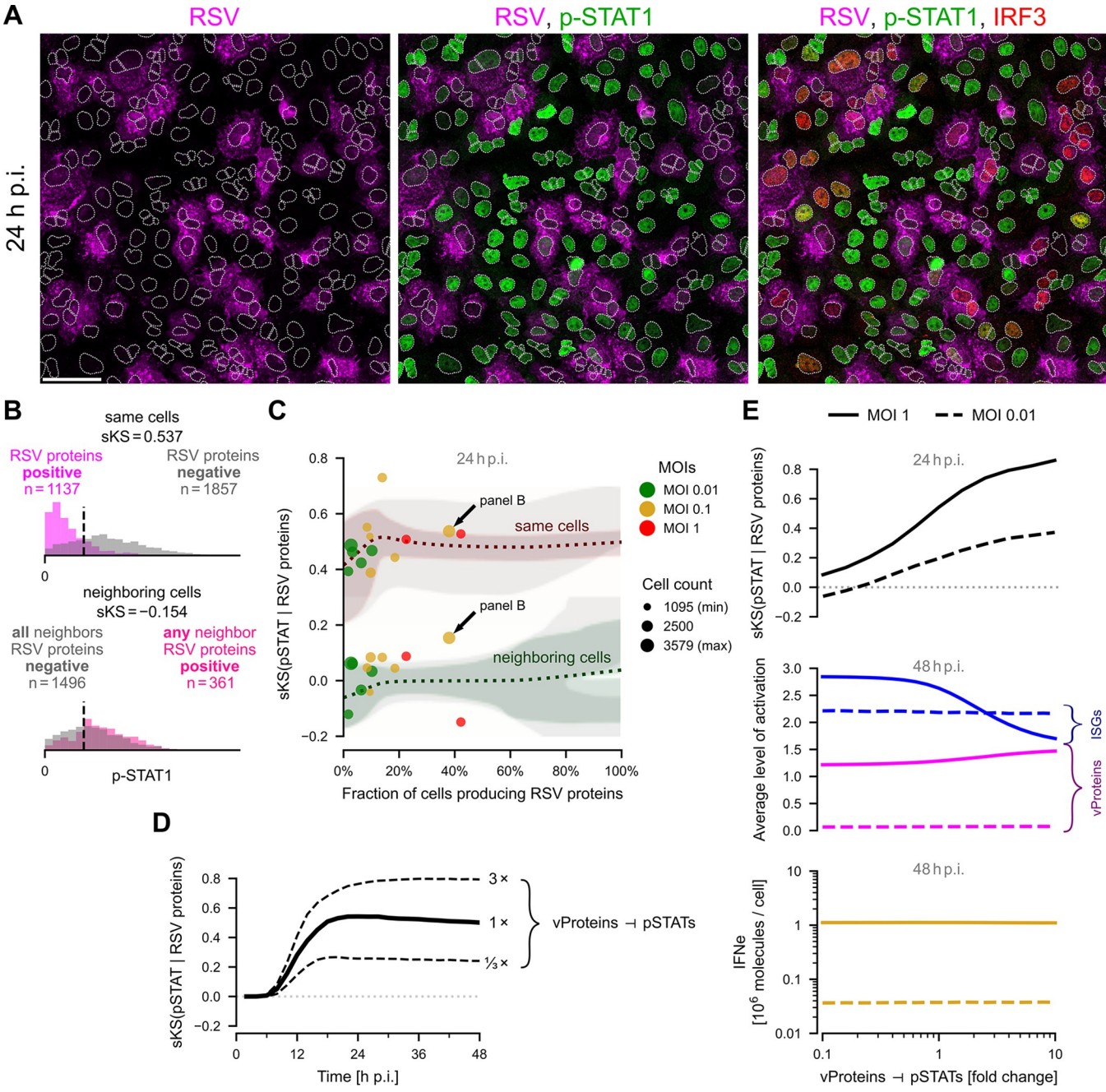

**Fig 3. Relation between RSV proteins and STAT signaling. A.** A549 cells 24 h post infection (p.i.) with RSV at an MOI of 0.1, immunostained for RSV proteins–magenta, p-STAT1 –green, and IRF3 –red; left to right, incremental overlays. White dotted lines are nuclear outlines determined based on DAPI counterstaining (channel not shown). Scale bar, 50 μm. **B.** Distributions of the mean nuclear p-STAT1 intensities in cells either expressing or not expressing RSV proteins (top subpanel, 'same cells') and in cells not expressing RSV proteins but either having or not a neighboring cell expressing RSV proteins (bottom subpanel, 'neighboring cells'). Pairs of histograms, cell counts, and signed Kolmogorov–Smirnov (sKS) statistics are given for a representative experiment of an MOI of 0.1 (marked in panel C with an arrow). **C.** Dependence of the 'same cells' sKS(pSTATs | RSV proteins) statistics and 'neighboring cells' sKS(pSTATs | RSV proteins) statistics on the fraction of RSV proteins-expressing cells at 24 h p.i. Marker sizes correspond to the number of cells quantified in each experiment. Experimental results are juxtaposed with model predictions. Colored dotted lines show mean predictions for default parameters, colored contours show 95% CrI computed for default parameters, and gray contours show 95% CrI computed for randomly perturbed default parameters with log-normal (σ = 0.2) noise. In both cases, CrI bands are computed based on multiple simulations each with $n$ = 2500 cells. **D.** 'Same cells' sKS(pSTATs | RSV proteins) over the period of two days post infection at an MOI of 0.1, according to the model. The dashed lines were obtained at different strengths of the vProteins ⊣ pSTATs inhibition. **E.** Model-based analysis of the influence of the vProteins ⊣ pSTATs inhibition: 'Same cells' sKS(pSTATs | RSV proteins) (upper subpanel, at 24 h p.i.), average ISGs and vProteins states (middle subpanel, at 48 h p.i.), average level of extracellular interferon IFNe (lower subpanel, at 48 h p.i.).

3D we show that the sKS statistic for 'same cells' increases in time from 0 to 0.5 (stabilizing at 24 h p.i.), which indicates more pronounced virus-induced pSTATs deactivation at later times. The model also predicts that the value of the 'same cells' sKS increases with the strength of the vProteins ⊣ pSTATs inhibition (Fig 3D and 3E), but is insensitive to variations in the strengths of the inhibition of pIRF3 and the inhibition of IFNi by vProteins (Fig J panel a in S1 Appendix). The 'neighboring cells' sKS statistics remain close to zero regardless of the strengths of all inhibitions originating from vProteins (Fig J panels a and b in S1 Appendix).

In agreement with the model, we observed that activation of STAT1/2 is key for attenuation of viral spread; prestimulation with IFNβ substantially lowered viral load (Fig E in S1 Appendix), while IFNAR or STAT1 or STAT2 KOs (Fig D panels c and a in S1 Appendix) increased viral load. Surprisingly, the model indicates that the strength of pSTATs inhibition by vProteins — although influencing the status of pSTATs in vProteins-expressing cells — very weakly influences the overall spread of infection (Fig 3E). For an MOI of 1 (but not for an MOI of 0.01) and stronger vProteins ⊣ pSTATs inhibition, the average status of ISGs is lowered, but this leads to a very modest increase of the average status of vProteins at 48 h p.i. and does not influence extracellular IFN accumulation. This is because, in individual cells, inhibition of pSTATs and consecutive depletion of the build-up of ISGs typically occurs after the (irreversible) accumulation of vProteins, and is thus not advantageous to the virus according to the model. This suggests that autocrine pSTATs activation does not play a significant role during RSV infection of A549 cells.

**RSV proteins inhibit IRF3 activation.** Viral RNA is required for synthesizing RSV proteins as well as for activating IRF3; these two processes, however, may but do not need to occur simultaneously in the same cell. Accordingly, in our immunostaining images, there are cells expressing RSV proteins with and without active IRF3, as well as cells with active IRF3 but devoid of RSV proteins (Fig 4A). Imaging data collected at different time points (16, 20, 24, 36, 40, and 48 h p.i.) and various initial MOIs (0.01, 0.1, and 1) show that approximately 40% of RSV proteins-expressing cells have active IRF3, with a good agreement with the model (Fig 4B). Conversely, approximately 50–80% of cells with active IRF3 express RSV proteins (Fig 4C). In experiments, the percentage of IRF3-active cells is somewhat larger for a larger fraction of cells expressing RSV proteins, but this finding is not reproduced by our model. An auxiliary demonstration of the vProteins ⊣ pIRF3 inhibition is provided in Fig G in S1 Appendix, where a sizable fraction of cells displays NF-κB but no IRF3 activity at 24 h and 40 h p.i. (but not at 16 h p.i.). In Fig 4B and 4C, we observe high variability between experimental replicates, indicating that the counterbalance between the strength of the innate immune response (indicated by the fraction of cells with active IRF3) and virus spread (indicated by the fraction of cells with RSV proteins) is heterogeneous and sensitive to experimental conditions. Analogously to Fig 3C, we plot 95% CrI in Fig 4B and 4C. These CrI show that the observed variability in experimental replicates could be attributed to relatively small variations in biochemical rates, which are expected between cell populations.

To show the importance of inhibitions, we include model predictions with the vProteins ⊣ pIRF3 inhibition and with all inhibitions either turned off or made 10-fold stronger (Fig 4B and 4C). This analysis highlights the key role of the vProteins ⊣ pIRF3 inhibition in the regulation of the proportions of cells co-expressing active IRF3 and RSV proteins with respect to cells expressing RSV proteins (Fig 4B) or active IRF3 (Fig 4C). According to the model, the fraction of cells that display both RSV proteins and IRF3 activity increases in time (as the infection progresses) and substantially decreases with the strength of inhibition of pIRF3 by vProteins (Fig 4D, left subpanel). Consequently, the overall fraction of IRF3-active cells is lower for stronger inhibition (Fig 4D, right subpanel).

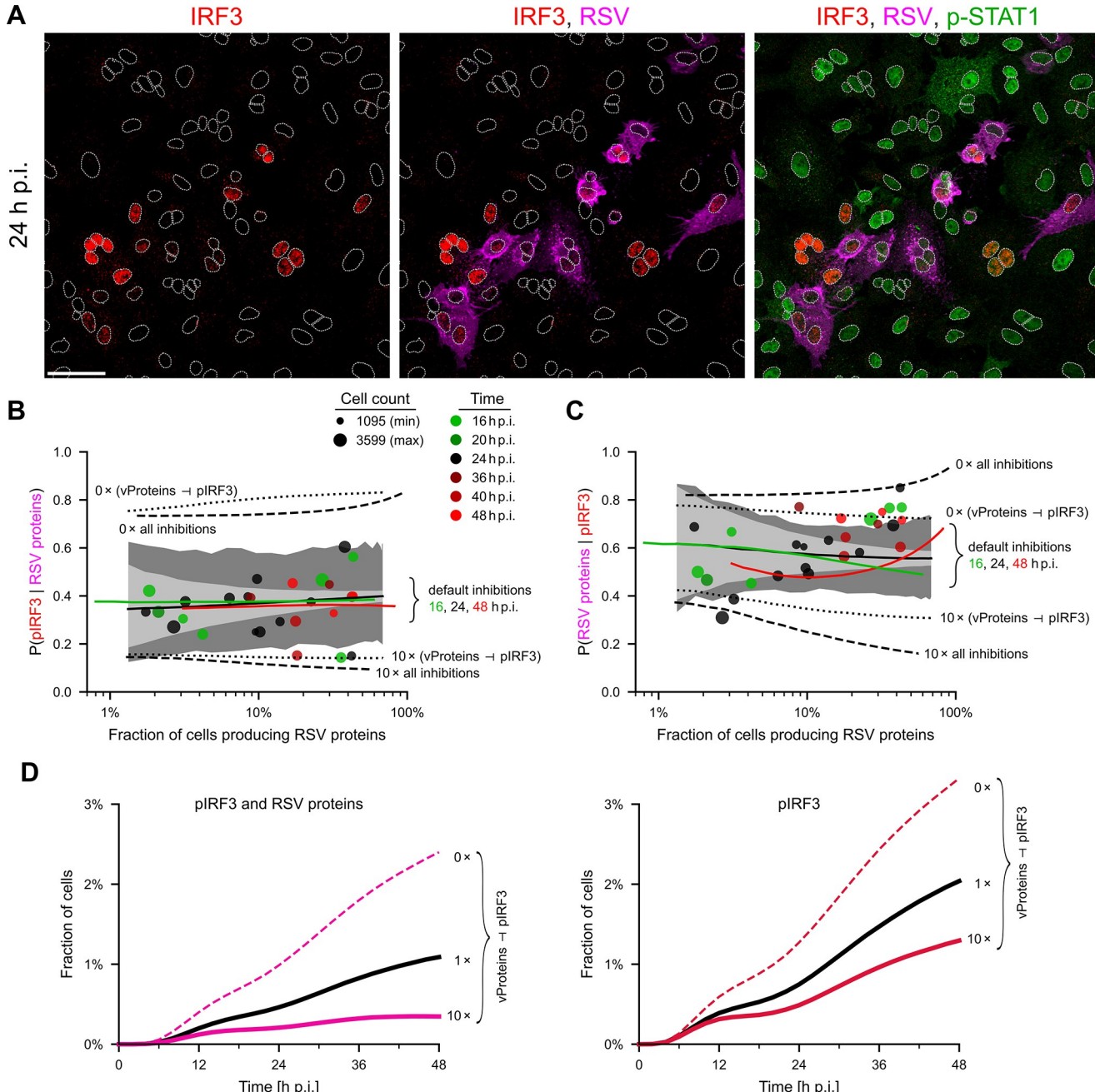

**Fig 4. Relation between expression of viral proteins and IRF3 activity. A.** Incremental overlays of images of A549 cells 24 h post infection (p.i.) with RSV at an MOI of 0.1, immunostained for IRF3 –red, RSV–magenta, and p-STAT1 –green. White dotted lines are nuclear outlines determined based on DAPI counterstaining (channel not shown). Scale bar, 50 μm. **B.** Conditional probability $P(\text{pIRF3} \mid \text{RSV proteins})$ as a function of the percentage of RSV proteins-expressing cells. Experiments (at 16, 20, 24, 36, 40, 48 h p.i.)–disks, model (at 16, 24, 48 h p.i.)–lines (line points were computed for a specific MOI, resulting in a given proportion of RSV-positive cells; results are averages over 1000 stochastic simulations on the $100 \times 100$ lattice). Dashed lines show model predictions (at 24 h p.i.) without all 5 inhibitions or with 10 times stronger inhibitions (as shown in the figure). Dotted lines show model predictions (at 24 h p.i.) without the `vProteins ⊣ IRF3` inhibition or with 10 times stronger inhibition. Light gray contours show 95% CrI computed for default parameters; dark gray contours show 95% CrI computed for randomly perturbed default parameters with log-normal ($\sigma = 0.2$) noise, both for $n = 2500$ cells. **C.** Conditional probability $p(\text{RSV proteins} \mid \text{pIRF3})$, notation as in panel B. **D.** Proportion of the (IRF3 & vProtein)-positive cells (left) and IRF3-positive cells (right) as a function of time for a default strength of the `vProteins ⊣ pIRF3` inhibition, no inhibition, and the inhibition 10 times stronger than the default.

**Mutual exclusion between the expression of RSV proteins and the synthesis of IFNβ.**
Viral RNA triggers production of IFN (via activation of IRF3) and is required for the synthesis of viral proteins. However, as we can see in immunostaining images in Fig 5A (as well as in Fig 2A and Fig H in S1 Appendix), the accumulation of IFN and RSV proteins rarely coincides in the same cell. The analysis of immunostaining images indicates that about 10% of cells expressing RSV proteins exhibit IFN accumulated during 2 hours of brefeldin A treatment that blocks IFN secretion (Fig 5B). In turn, 30–40% of the cells exhibiting accumulated IFN have RSV proteins (Fig 5C). Both percentages do not change significantly with initial MOIs (0.01, 0.1, and 1) and over time (experimental time points: 16, 20, 24, 36, 40, and 48 h p.i.), and are in good agreement with the model. As previously demonstrated in Figs 4B, 4C, 5B, and 5C we observe high variability between experimental replicates, which again can be attributed to stochastic noise and relatively small variability of biochemical parameters.

When comparing predictions of the model in which inhibitions `vProteins ⊣ pIRF3` and `vProteins ⊣ IFNi` are either turned off or made 10-fold stronger (dotted lines in

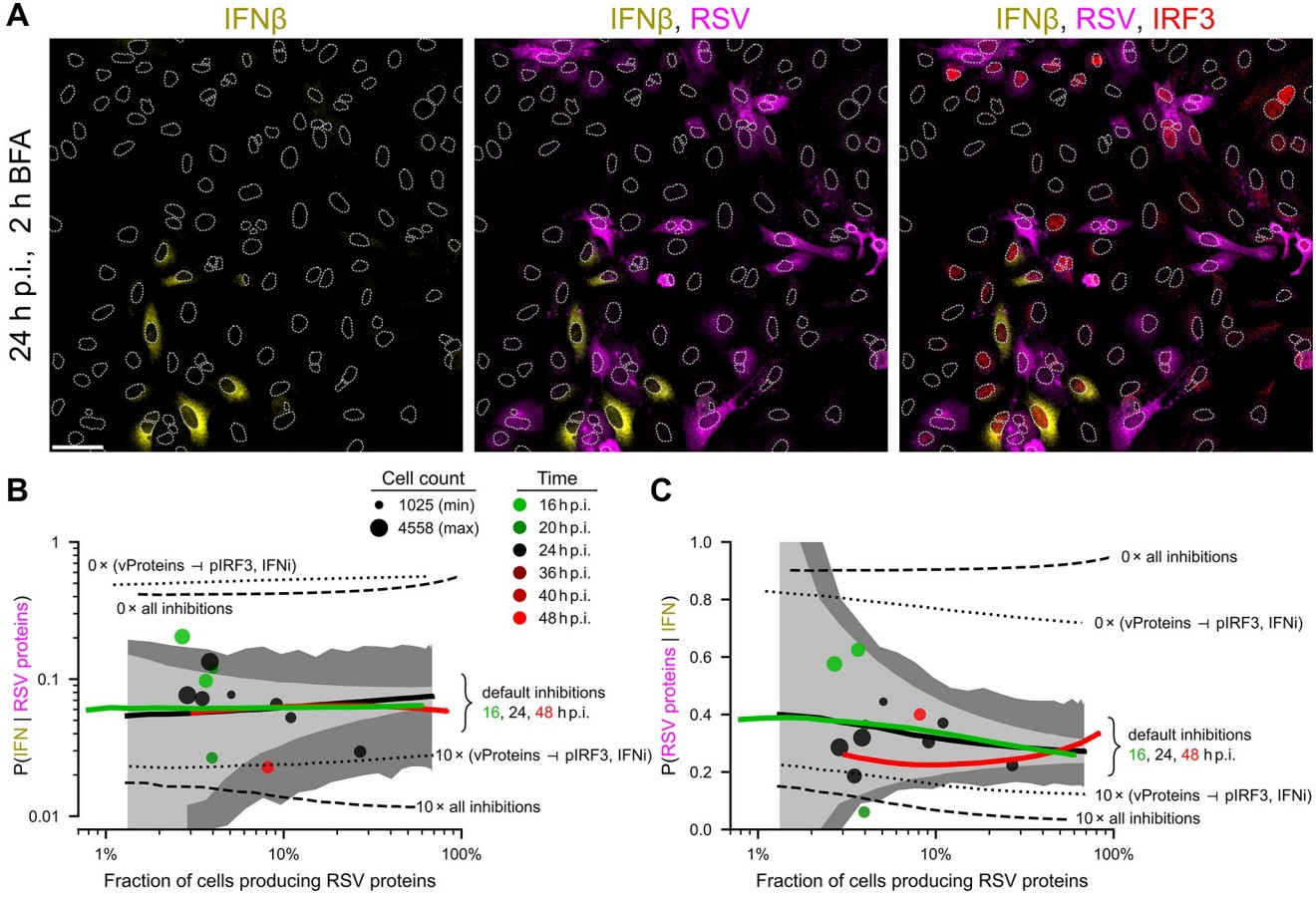

**Fig 5. Mutually exclusive expression of RSV proteins and synthesis of IFNβ. A.** Incremental overlays of images of A549 cells 26 hours post infection (h p.i.) with RSV at MOI 0.1, immunostained for IFNβ –yellow, RSV–magenta, and IRF3 –red. To improve detection of intracellular IFNβ, we used a 2-hour-long treatment with brefeldin A (BFA) before cell fixing, which prevented secretion of IFNβ produced during this treatment. White dotted lines are nuclear outlines determined based on DAPI counterstaining (channel not shown). Scale bar, 50 μm. **B.** Conditional probability $P$(IFN | RSV proteins) as a function of the percentage of RSV proteins-expressing cells. Experiments (at 16, 20, 24, 36, 40, 48 h p.i.)–disks, model (at 16, 24, 48 h p.i.)–lines (line points were computed for a specific MOI, resulting in a given proportion of RSV-positive cells; results are averages over 1000 stochastic simulations on the 100 × 100 lattice). Dashed lines show model predictions (at 24 h p.i.) without all 5 inhibitions or with 10 times stronger inhibitions (as shown in the figure). Dotted lines show model predictions (at 24 h p.i.) without `vProteins ⊣ pIRF3` and `vProteins ⊣ IFNi` inhibitions, or with these two inhibitions 10 times stronger. Light gray contours show 95% CrI computed for default parameters; dark gray contours show 95% CrI computed for randomly perturbed default parameters with log-normal (σ = 0.2) noise, both for $n$ = 2500 cells. **C.** Conditional probability $p$(RSV proteins | IFN), notation as in panel B.

Fig 5B and 5C) to predictions of the model in which all inhibitions are either turned off or made 10-fold stronger (dashed lines in Fig 5B and 5C), we can see the crucial role of inhibitions `vProteins ⊣ pIRF3` and `vProteins ⊣ IFNi` in achieving correct proportions of cells co-expressing IFN and RSV proteins with respect to cells expressing RSV proteins (Fig 5B) or IFN (Fig 5C).

**Model-based analysis of the influence of inhibitory interactions on the progression of viral infection.** The effect of antagonistic inhibitions from `vProteins` to `pIRF3` and `IFNi`, and from `ISGs` to `vRNA` and `vProteins` (Fig 6A) on the progression of the infection is further studied using the validated model. We focus on the case of MOI = 0.01, in which the immune response is granted time to develop.

As shown in Fig 6B, the fraction of cells producing both IFN and RSV proteins grows in time as the infection spreads, but remains below 0.2% even at 48 h p.i. This fraction is controlled by the strengths of inhibitions `vProteins ⊣ pIRF3` and `vProteins ⊣ IFNi`. When these inhibitory interactions are 10 times stronger, the fraction is close to 0, whereas their removal increases the fraction fivefold, to 1%. Correspondingly, the removal of these inhibitory interactions increases the overall fraction of cells producing IFN (at 48 h p.i.) more than twofold, to above 1% (Fig 6C). However, when 10-fold stronger inhibitory interactions are assumed, the relative effect on the fraction of IFN-producing cells is negligible, since only about 10% of cells producing IFN have `vProteins` (consistently with data in Fig 5B). The effect of removing inhibitory interactions `vProteins ⊣ pIRF3` and `vProteins ⊣ IFNi` is forwarded through subsequent tiers of the immune response leading to an increase in the levels of `pSTATs` (Fig 6D) and `ISGs` (Fig 6E). The most significant increase is observed, respectively, at about 20 and 30 h p.i. Consequently, induction of a more pronounced antiviral state at about 30 hours (epitomized by an increased level of `ISGs`) leads to a decrease of `vProteins`-producing cells after that time point (Fig 6F). In summary, the analysis presented in Fig 6B–6F dissects regulatory steps in which the virus enhances its propagation by suppressing the innate immune response.

Next, we analyze how the strengths of inhibitory interactions mediated by the immune response, `ISGs ⊣ vProteins` and `ISGs ⊣ vRNA`, influence the propagation of infection and activation of the immune response. As shown in Fig 6G, an increase in the strength of the `ISGs ⊣ vProteins` inhibition leads to a decrease in the fraction of `vProteins`-producing cells and a simultaneous increase in the number of interferon-producing cells. Clearly, this inhibitory interaction attenuates viral infection and promotes interferon signaling. The effect of inhibitory interaction `ISGs ⊣ vRNA` is different; an increase in its strength leads to a moderate decrease of the fraction of `vProteins`-producing cells, but also to a substantial decrease of interferon-producing cells (Fig 6H). This is because viral RNA is necessary for both the synthesis of viral proteins and activation of the IRF3 → IFN pathway.

## Influence of defective viral genomes (DVGs) on virus spread

In the model we have not accounted for the potential formation of defective viral particles with incomplete genomes that are capable of activating IRF3 (triggering IFN synthesis), but are incapable of producing infectious virions and may not be able to express proteins that would inhibit IFN synthesis. In experimental conditions, DVGs may emerge as an artifact during viral amplification [38] and although we cannot rule out their presence in our experiments, the obtained results can be explained by the model that neglects their presence. Since DVGs are known to be important drivers of the immune response [17], we analyze theoretically the consequence of their presence on IFN synthesis and virus spread.

For the sake of simplicity we assume that there is a fraction of cells infected solely with DVGs ($f_{DVG}$) that are capable of activating IRF3 but, in the absence of replication-competent

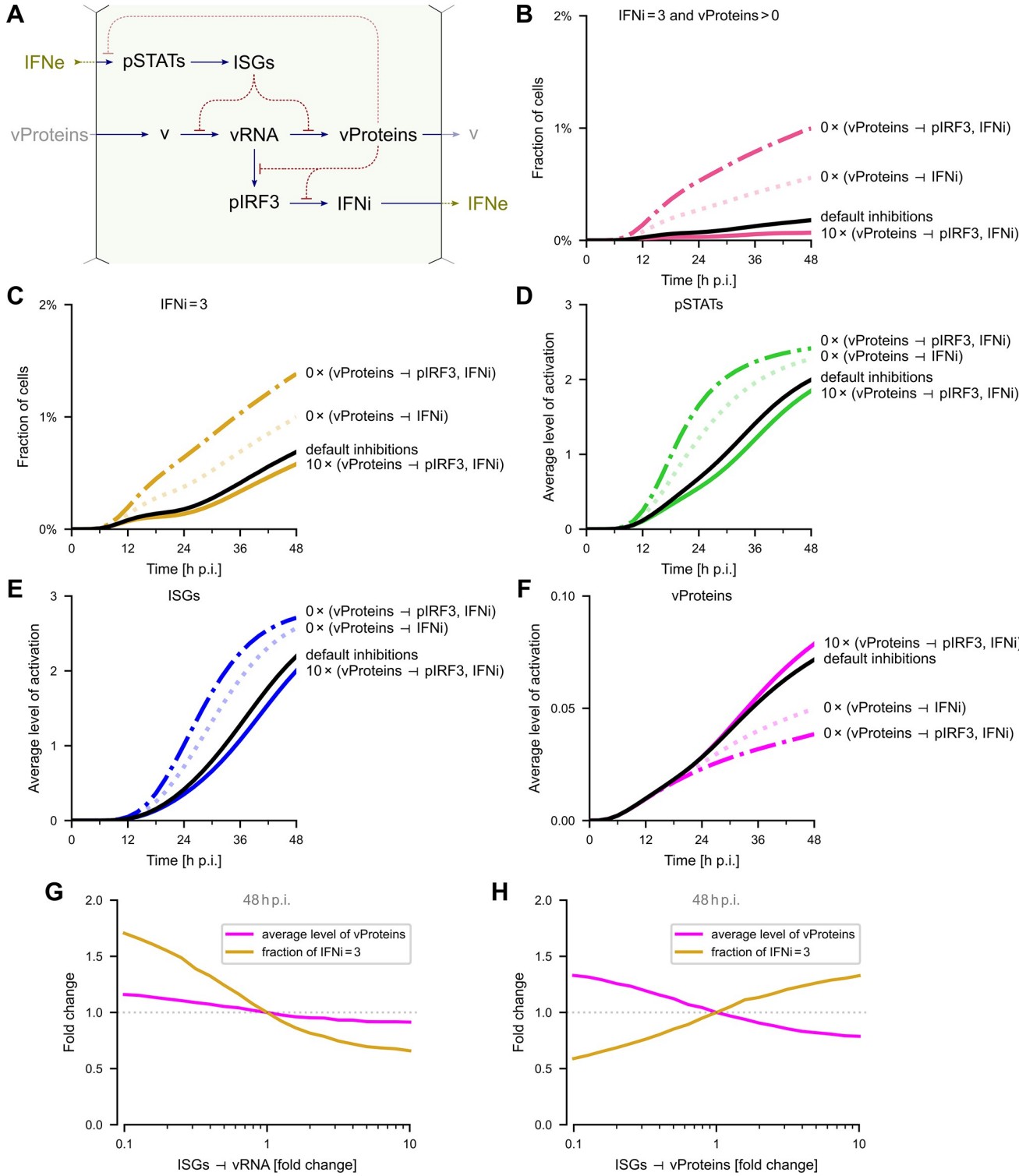

**Fig 6. Influence of inhibitions on the progression of viral infection (model). A.** Simplified scheme of antagonistic interactions present in the model. **B–E.** (**B**) Fraction of cells producing `IFNe` (`IFNi = 3`) and `vProteins` (`vProteins > 0`) simultaneously, (**C**) fraction of cells producing `IFNe`, (**D**) average activation of `pSTATs`, (**E**) average activation of `ISGs`, (**F**) fraction of cells producing `vProteins`; all as a function of time for an MOI of 0.01 for default inhibition strengths (black line), inhibitions `vProteins ⊣ IRF3` and `vProteins ⊣ IFNi` 10 times stronger (thick line), and turned off (dashed, dotted and dash-dotted lines). **G.** Change of the average level of `vProteins` and of the fraction of cells producing `IFNe` as a function of the `ISGs ⊣ vProteins` inhibition, at 48 h p.i. at an MOI of 0.01. **H.** Change of the average level of `vProteins` and of the fraction of cells producing `IFNe` as a function of the `ISGs ⊣ vRNA` inhibition, at 48 h p.i. at an MOI of 0.01.

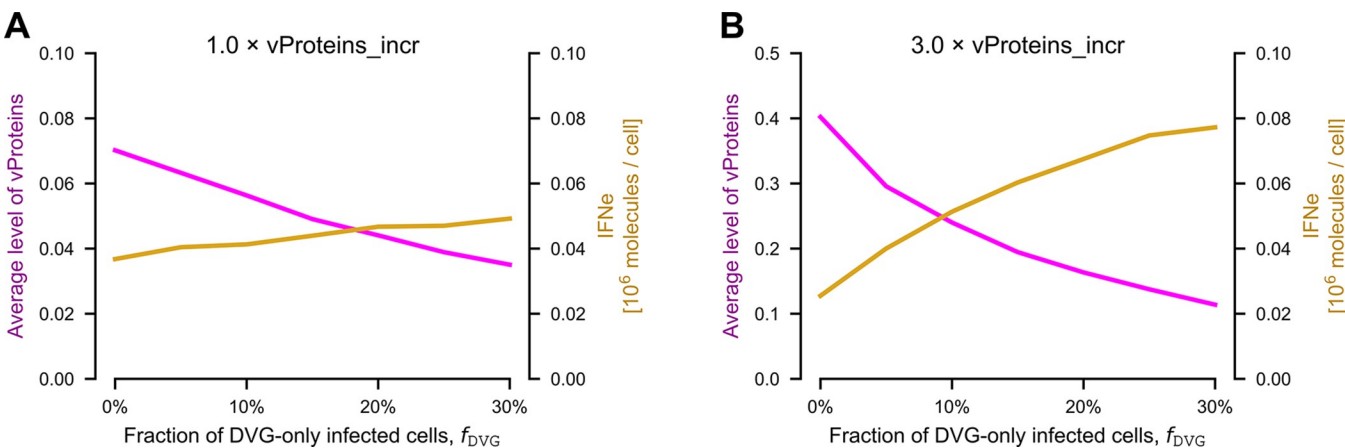

**Fig 7. Influence of DVGs on IFN expression and viral progeny (model). A**. The average status of `vProteins` and `IFNe` as a function of the percentage of cells infected solely by DVGs at 48 h p.i. Initial MOI: 0.01. **B**. Same as in panel A, but with the `vProteins` forward rate coefficient (`vProteins_inc`) increased 3-fold. Note the different range on the left vertical axis.

helper virions, cannot express viral proteins. In Fig 7A we show a relatively modest increase of `IFNe` and about a twofold decrease in the average `vProteins` level as $f_{DVG}$ increases from 0 to 30%. This surprisingly weak influence is a consequence of the fact that for the default model parameters, cells are able to express IFN before viral proteins are produced. However, when the `vProteins` forward rate is increased threefold, the effect of DVGs on infected cells is much stronger, leading to a fourfold reduction of the level of `vProteins` (Fig 7B). Overall, this analysis indicates that DVGs can be critical drivers of innate responses, but only in the case of viruses that are capable of quickly terminating the synthesis of IFN.

## Discussion

Viral infection, when viewed at the cell-population level, leads to both the multiplication of viral particles and the triggering of the innate immune response. The intuitively expected and experimentally widely observed antagonism between these two processes [29] can be investigated at the single-cell level. Based on cell-population and single-cell experiments involving RSV, we constructed and constrained a stochastic, agent-based model that elucidates this mutual antagonism.

To assemble the model, we confirmed experimentally the importance of the RIG-I/MAVS → IRF3 → IFNβ → IFNAR → STAT1/2 → ISGs axis in antiviral signaling by characterizing the effects of knockouts of its components at the level of (epithelial) cell population. The lack of RIG-I, MAVS, IRF3, IFNAR, STAT1, or STAT2 leads to at least several-fold increase in viral load, in agreement with earlier studies. Seth et al. [39] showed that knockdowns of MAVS and RIG-I reduce IRF3 phosphorylation and promote viral spread (a direct interaction between MAVS and IRF3 in virus-infected cells was demonstrated later by Liu et al. [40]). Wathelet et al. [41] showed that in response to virus infection, IRF3 (as well as IRF7), jointly with ATF-2/c-Jun and NF-κB, form the enhanceosome to trigger IFN expression. Correspondingly, Hatesuer et al. [42] showed that the lack of both IRF3 and IRF7 resulted in the absence of type I interferons after IAV infection in mice and resulted in a strong increase of mortality. In the case of RSV infection, IFNβ → IFNAR signaling plays a dominant role, while in the case of IAV, IFNλ → IFNLR is of equal importance. As shown by Goritzka et al. [43], IFNAR1-deficient mice displayed increased lung viral load during RSV infection. Correspondingly, Crotta et al. [44] showed that a simultaneous deletion of IFNAR and IFNLR renders mice vulnerable

to IAV infection of the respiratory tract. Both STAT2- and STAT1-deficient mice are more vulnerable to viral infections, however, unsurprisingly, the effect is more pronounced for STAT1-deficient mice as they also cannot respond to type II interferons, reviewed by Meyts & Casanova [45].

The proposed model recapitulates the expected time to production of IFN by early responders, which is in the range of 4–8 h [2,46]. The low fraction of IFN-producing cells is also on par with other estimates indicating that up to a few percent of cells produce interferons, see, e.g., [6,47,48]. As in experiments, the scant sources of IFN, especially at a low MOI, amplify cell population heterogeneity by giving rise to characteristic foci of IFN-activated cells [46].

The model allowed us to investigate correlations between the levels of key proteins in the same cell as well as in adjacent cells, which enabled quantitative comparison with spatial patterns observed in immunostaining images. Both the experimental data and the model indicate that while viral RNA triggers the innate immune response, viral proteins inhibit this response at 3 levels: by blocking activation of IRF3, by blocking the synthesis of IFN, and by inhibiting STATs activation. The first two interactions promote the spread of infection, whereas, as suggested by the model, the last one has a minimal influence on virus spread. The fact that the direct inhibition of STAT phosphorylation by RSV (nonstructural) proteins has little influence on RSV replication in the model suggests that this interaction may be implicated in regulatory processes not included in the model. It is plausible that disabling the production of ISG-encoded proteins redirects ribosomes to the production of viral proteins, enabling efficient generation of viral progeny. Consistent with this hypothesis, RSV has been shown to take over the global control of host cell translation [49]. The innate immune response inhibits viral propagation through paracrine IFN signaling, which activates STAT signaling which, in turn, upregulates expression of ISGs. Once a cell becomes infected, ISGs slow down the synthesis of RSV proteins, giving more time for activation of IRF3 and the synthesis of IFN. However, because the process of RSV protein synthesis is irreversible, at some point, RSV proteins accumulate and terminate both the activity of IRF3 and synthesis of IFN. A consequence of the `vProteins ⊣ pIRF3` and `vProteins ⊣ IFNi` inhibitions is that only about 40% of viral proteins-expressing cells have active IRF3 and as little as 10% of such cells produce IFN.

This picture is changed in the presence of DVGs; the model indicates that the fraction of cells infected solely by DVGs can, due to the lack of viral protein expression, serve as a source of IFN and in that way limit virus spread. The effect is modest for default model parameters but significant when faster accumulation of viral proteins is assumed — this implies that DVGs can be critical drivers of interferon-mediated responses, as shown experimentally by Sun et al. [38], but only in the case of viruses that would effectively inhibit interferon synthesis.

In many viral infections, an important factor limiting virus spread is a programmed death of infected cells [50,51]. Although RSV protein F can trigger apoptosis [52], two non-structural proteins of RSV, NS1 and NS2 effectively block apoptosis [53]. This blockage may be only transient, but in our immunostaining images we see that for at least 48 h p.i. the foci of infected regions remain confluent, suggesting an insignificant apoptotic rate. The protective effect may be at least in part attributed to the observed sustained nuclear residence of plausibly anti-apoptotic NF-κB. Since our model is calibrated for RSV based on our experimental data, it does not include infected cell death. As a consequence, the virus spread can be only slowed down (but not contained) by interferon signaling that mounts an antiviral state in not yet infected, bystander cells. Deaths of infected cells, if occurring during the early onset of infection, would change this picture through the elimination of infected cells before they produce or release new virions, as is the case during some other viral infections [54].

Finally, we note that IRF3 (independently of NF-κB) induces the expression of several genes—including ISG56/IFIT1, ISG54/IFIT2, and ISG60/IFIT3—in an IFN-independent

manner [55]. IFIT1/2/3 are involved in the inhibition of RSV infection [56], whereas RSV NS1 impinges on the expression of IFIT2/3 [13,57]. These antagonistic interactions are not included in the model. NF-κB, which is also activated in response to viral RNA, activates the transcription of proinflammatory interleukins IL-6 and IL-8 [58]. Their primary role is the regulation and attraction of immune cells, which are however absent both in our experiments and in our model.

In summary, single-cell experimental data and the proposed stochastic, agent-based model demonstrate the importance of reciprocally inhibitory interactions between the virus and the innate immune response. Because of the irreversible progression of virus replication (not limited by cell death), these interactions do not lead to typical bistability; instead, they result in divergent single-cell trajectories. A single infected cell may either rapidly initiate a cascade of new infections or become a transient source of IFN, allowing neighboring cells to enter an antiviral state and slow down virus spread. As a consequence, when the respiratory epithelium is infected by a small number of virions, heterogeneity of single-cell responses introduced by noise and mutually inhibiting interactions may lead to different outcomes of infection at the organismal level.

## Methods

### Experimental methods

**Cell lines and cell culture.**   The A549, HeLa (used for RSV proliferation), and HEK-293T (used for the production of lentiviral particles) cell lines were purchased from ATCC. A549 cells were cultured in F-12K basal medium. HeLa and HEK-293T cells were maintained in Dulbecco's modified Eagle's medium (DMEM). Both media were supplemented with 10% FBS and a penicillin–streptomycin antibiotic solution. Cells were cultured under standard conditions (37℃ with 5% $CO_2$) in a Series 8000 WJ incubator (Thermo Fisher Scientific) and kept in monolayers up to 90% confluency.

For STAT1 and STAT2 gene knockouts, a CRISPR lentiviral vector system was used. Commercially available sgRNAs sets were purchased from Applied Biological Materials (catalog numbers, K0002501 and K2300601, respectively). STAT1 knockout was obtained by targeting sequence TGATCCAAGCAAGCATTGGG in exon 9 of STAT1 gene; STAT2 knockout was obtained by targeting sequence AGCTCCCATTGACCACGGGT in exon 8 of STAT2 gene. Two days after transfection of HEK-293T cells with plasmids encoding Cas9 nuclease (catalog number K002; Abm), lentivirus packaging particles (catalog number LV053; Abm), and mentioned earlier of each single sgRNA, the media with lentiviral particles were collected, enriched with 8 mg/mL Polybrene, and filtered through 0.45-mm syringe filters. Next, each lentiviral supernatant was used to transduce A549 wild-type cells, which were subcultured at low confluence (seeding density of $5\times10^4$ cells per 30-mm dish). After another 2 days, A549 cells were subjected to selection with 800 mg/mL of G418 for 10 days and seeded into a 96-well plate to obtain single-cell colonies. Clones were validated by Western blotting of p-STAT1 and p-STAT2 in response to IFNβ. Additionally, the KO clones selected for further experiments were verified by sequencing.

RIG-I KO cells were purchased from InvivoGen (catalog number a549d-korigi). The A549 IRF3 KO and MAVS KO cell lines were obtained as described by Korwek et al. [7]. The A549 IFNAR1 KO, IFNLR1 KO, IFNAR1 and IFNLR1 double KO cell lines were obtained as described by Czerkies et al. [28].

**RSV amplification and isolation.**   Respiratory syncytial virus strain A2 was purchased from ATCC and amplified in HeLa cells. The cells were seeded into 225-cm$^2$ tissue culture flasks (Falcon) and cultured as described above for 2 to 3 days until they reached 90%

confluence. The virus growth medium (DMEM with 2% FBS) and a virus dilution (with a target MOI of ~0.01) were prepared on the day of infection. The culture medium was removed and cells were washed once with phosphate-buffered saline (PBS) and overlaid with 10 mL of the inoculum. The virus was allowed to adsorb to cells for 2 h at 37℃, with occasional stirring. Next, additional virus growth medium was added to a total volume of 40 mL per flask. Infected cells were cultured for 3–6 days at 37℃ until the development of cytopathic effects was observed in at least 80% of the cells. Virus-containing culture fluid was then collected and clarified by centrifugation at 3,000 *g* at 4℃ for 20 min. Next, virus particles were precipitated by adding 50% (wt/vol) polyethylene glycol 6000 (PEG 6000) (Sigma-Aldrich) in NT buffer (150 mM NaCl, 50 mM Tris-HCl [pH 7.5]) to a final concentration of 10% and stirring the mixture gently for 90 min at 4℃. Virus was centrifuged at 3,250 *g* for 20 min at 4℃ and re-centrifuged after the removal of the supernatant to remove the remaining fluid. The pellet was suspended in 1 mL 20% sucrose in the NT buffer, aliquoted, and stored at −80℃.

**RSV quantification.** The virus concentration was quantified using an immunofluorescence protocol. HeLa cells were seeded onto microscopic coverslips and cultured upon reaching 90 to 100% confluence. Serial dilutions of virus samples were prepared in the virus growth medium in a range of $10^{-3}$ to $10^{-7}$. After washing with PBS, cells were overlaid with the diluted virus (for each virus dilution, two coverslips were used). The virus was allowed to adhere for 2 h and occasionally stirred. Afterward, the virus-containing medium was removed, and cells were overlaid with fresh virus growth medium and cultured for 24 h. Next, cells were washed with PBS and fixed with 4% formaldehyde for 20 min at room temperature. Cells were stained using a standard immunofluorescence protocol with an anti-RSV fusion glycoprotein antibody (catalog number ab43812; Abcam). Cells containing stained viral proteins were counted using a Leica TCS SP5 X confocal microscope. The virus concentration was calculated using the following formula: (average number of infected cells)/(dilution factor × volume containing virus added) = infectious particles/mL.

**Compounds and stimulation protocols.** Human IFNβ 1a was purchased from Thermo Fisher Scientific (catalog number PHC4244) and prepared according to the manufacturer's instructions. For cell stimulation, interferons were further diluted to the desired concentration in F-12K medium supplemented with 2% FBS. The decreased FBS content was used to prevent the inhibition of viral attachment and entry at the second stage of experiments. For interferon-and-virus experiments, the cell culture media were exchanged for interferon-containing or control media at time zero and were not removed until the end of the experiment. Appropriately diluted virus was added in small volumes (less than 10 μl) directly into the wells. The even distribution of the virus across the cell population was aided by the intermittent rocking of the plate for 2 h. For intracellular IFNβ visualization, a brefeldin A solution (catalog number 00-4506-51; Invitrogen) was added 2 h prior to cell fixation.

**Western blotting.** At the indicated time points, cells were washed twice with PBS, lysed in Laemmli sample buffer containing dithiothreitol (DTT), and boiled for 10 min at 95℃. Equal amounts of each protein sample were separated on 4% to 16% Mini-Protean TGX stain-free precast gels using the Mini-Protean Tetra cell electrophoresis system (Bio-Rad). Upon the completion of electrophoresis, proteins were transferred to a nitrocellulose membrane using wet electrotransfer in the Mini-Protean apparatus (400 mA for 50 min). The membrane was rinsed with TBST (Tris-buffered saline [TBS] containing 0.1% Tween 20; catalog number P7949, Sigma-Aldrich) and blocked for 1 h with 5% BSA–TBS or 5% nonfat dry milk. Subsequently, the membranes were incubated with one of the primary antibodies diluted in a 5% BSA–TBS buffer at 4℃ overnight. After thorough washing with TBST, the membranes were incubated with secondary antibodies conjugated to a specific fluorochrome (DyLight 800; Thermo Fisher Scientific) or horseradish peroxidase (HRP-conjugated polyclonal anti-mouse/

anti-rabbit immunoglobulins; Dako) diluted 1:5,000 in 5% non-fat dry milk–TBST for 1 h at room temperature. The chemiluminescence reaction was developed with the Clarity Western ECL system (Bio-Rad). For GAPDH detection, hFAB rhodamine anti-GAPDH primary antibody (Bio-Rad) was used. Specific protein bands were detected using the ChemiDoc MP imaging system (Bio-Rad).

**Immunostaining.** For staining of intracellular proteins, cells were seeded onto 12-mm round glass coverslips, which were previously washed in 60% ethanol–40% HCl, thoroughly rinsed with water, and sterilized. After stimulation, cells on the coverslips were washed with PBS and immediately fixed with 4% formaldehyde (20 min at room temperature). Cells were then washed 3 times with PBS and incubated with 100% cold methanol for 20 min at −20℃. After washing with PBS, coverslips with cells were blocked and permeabilized for 1.5 h with 5% BSA (Sigma-Aldrich) with 0.3% Triton X-100 (Sigma-Aldrich) in PBS at room temperature. Subsequently, coverslips with cells were incubated with primary antibodies diluted in a blocking solution overnight at 4℃. Cells were then washed 5 times with PBS, and then appropriate secondary antibodies conjugated with fluorescent dyes were added in a blocking solution for 1 h at room temperature. Subsequently, cells were washed with PBS and their nuclei were stained with 200 ng/mL 4',6-diamidino-2-phenylindole (DAPI; Sigma-Aldrich) for 10 min. After a final wash in MilliQ water, coverslips with stained cells were mounted onto microscope slides with a Vectashield Vibrance Antifade Mounting Medium (Vector Laboratories). Multichannel fluorescence images were acquired with a Leica TCS SP5 X confocal microscope.

**Antibodies.** *Antibodies for Western blotting.* **Primary antibodies**: anti-phospho-STAT1 (Tyr701) (clone 58D6) (catalog number 9167; Cell Signaling Technologies; 1:1000); anti-phospho-STAT2 (Tyr690) (clone D3P2P) (catalog number 88410; Cell Signaling Technologies; 1:1000); anti-phospho-IRF3 (Ser396) (clone 4D4G) (catalog number 4947; Cell Signaling Technologies; 1:1000); anti-IRF3 (clone D6I4C) (catalog number 11904; Cell Signaling Technologies; 1:1000); anti-RIG-I (clone D14G6) (catalog number 3743; Cell Signaling Technologies; 1:1000); anti-STAT1 (catalog number 610116; BD Biosciences; 1:1000), anti-STAT2 (catalog number PAF-ST2; R&D Systems; 1:1000); anti-PKR (clone B-10) (catalog number sc-6282; Santa Cruz Biotechnology; 1:1000); anti-RNase L (clone E-9) (catalog number sc-74405; Santa Cruz Biotechnology; 1:1000); anti-OAS1 (clone F-3) (catalog number sc-374656; Santa Cruz Biotechnology; 1:1000), anti-Respiratory Syncytial Virus (clone 2F7) (catalog number ab43812; Abcam; 1:1000); hFAB Rhodamine anti-GAPDH (catalog number 12004168; Bio-Rad; 1:10,000).

**Secondary antibodies:** goat anti-rabbit IgG (H+L), DyLight 800 4X PEG (catalog number SA5-35571; Thermo Fisher Scientific; 1:10,000); goat anti-mouse IgG (H+L), DyLight 800 4X PEG (catalog number SA5-35521; Thermo Fisher Scientific; 1:10,000); StarBright Blue 700 goat anti-mouse IgG (catalog number 12004159; Bio-Rad; 1:10,000), rabbit anti-goat immunoglobulins/HRP (catalog number P0449; Agilent; 1:10,000).

*Antibodies for immunostaining.* **Primary antibodies**: anti-phospho-STAT1 Tyr701 (clone 58D6) (catalog number 9167; Cell Signaling Technologies; 1:1000); anti-IRF-3 (clone D-3) (catalog number sc-376455; Santa Cruz Biotechnology; 1:500), anti-Respiratory Syncytial Virus (catalog number ab20745; Abcam; 1:1000), anti-human interferon beta (catalog number MAB8142; R&D Systems; 1:100).

**Secondary antibodies**: donkey anti-rabbit IgG (H+L), Alexa Fluor 488 conjugate (catalog number A-21206; Thermo Fisher Scientific; 1:1000); donkey anti-mouse IgG (H+L), Alexa Fluor 555 conjugate (catalog number A-31570; Thermo Fisher Scientific; 1:1000); donkey anti-goat IgG (H+L), Alexa Fluor 633 conjugate (catalog number A-21082; Thermo Fisher Scientific; 1:1000).

## Computational methods and modeling

**Image quantification.** Confocal images obtained from immunostaining were analyzed using our in-house software (https://pmbm.ippt.pan.pl/software/shuttletracker). Nuclear regions were detected based on DAPI staining. The nuclei that were partially out of frame or mitotic were excluded from the analysis; outlines of overlapping nuclei were split based on geometric convexity defects when possible. To characterize cells with respect to the levels of RSV proteins, nuclear IRF3 (as a proxy of p-IRF3), accumulated IFNβ, and p-STAT1, we used image features that were visually checked and confirmed to adequately capture the immuno-fluorescence signal in a number of fields of view in several experiments. Specifically, cells were classified as: RSV proteins positive/negative—based on both the mean intensity of pixels of a perinuclear ring (after rejecting ⅓ of the brightest and ⅓ of the darkest pixels) and the sum of pixel intensities in the nuclear region and in the sigmoidally weighted nuclear halo around the nucleus in its corresponding Voronoi tile (2 features); p-IRF3 positive/negative—based on both the mean intensity of the nuclear region (after rejecting 10% brightest and 10% darkest pixels) and nuclear dominance (calculated as the difference between the mean nuclear and the mean perinuclear ring intensity normalized by the sum of these mean intensities, all calculated after rejecting 10% brightest and 10% darkest pixels; 2 features); IFNβ positive/negative—based on both the mean intensity of the 50% brightest pixels in the perinuclear ring and the sum of pixel intensities in the nuclear region and in the sigmoidally weighted nuclear halo around the nucleus in its corresponding Voronoi tile (2 features); p-STAT1 was quantified as the mean intensity of the nuclear region (after rejecting 10% brightest and 10% darkest pixels; 1 feature, not subjected to thresholding). Thresholds used to binarize the features of RSV proteins, IRF3, and IFNβ were selected to correctly identify cells visually identified as RSV proteins, p-IRF3, and IFNβ positive/negative in a number of fields of view in several experiments.

To express the extent to which the two distributions are disjoint, we calculated the signed Kolmogorov–Smirnov (sKS) statistic. The absolute value of sKS is that of the standard Kolmogorov–Smirnov statistics, whereas the sign additionally informs about the relative locations of two (empirical) probability densities (shown in histograms as probability density functions, PDFs). We used the notation sKS($X \mid Y$) to express the sKS statistics for a pair of distributions of $X$ (both continuous in the case of p-STAT1 or both binary otherwise), one obtained for $Y$-negative cells and the other obtained for $Y$-positive cells. The sKS statistic attains the values of $\pm 1$ for fully disjointed distributions and 0 for exactly overlapping distributions.

**Model justification and choice of parameters.** The model structure is justified and model parameters selected based on a series of experiments involving IFNβ stimulation and RSV infection of A549 cells, discussed in the following sections. Overall, parameter values were initially constrained based on cell-population experiments (Figs A–E in S1 Appendix), and then further manually adjusted based on single-cell data (Figs 3C, 4BC, and 5BC). Finally, we tuned feedback strengths based on IFNβ pre-stimulation experiments (Figs D, E in S1 Appendix) and single-cell data. Although the model parameters are constrained based on ample data collected in Figs A–E in S1 Appendix and Figs 3–5, our choice of parameters may not be unique [59,60].

When comparing model predictions with experimental data in Figs 3C, 4B, 4C, 5B and 5C we additionally show two types of 95% credibility intervals (95% CrI). First, to estimate the effects of stochastic noise, we run the model 10,000 times for default parameter values on a $50 \times 50$ lattice containing $n = 2500$ cells. Second, to estimate the combined effect of stochastic noise and parameter variation, we run the model for log-normally ($\sigma = 0.2$) perturbed parameters, also on a $50 \times 50$ lattice. We observe that when two types of noise are taken into account, variability between experimental and numerical replicates is similar.

In Figs A–E in S1 Appendix, overviewed below, we compare population-averaged model trajectories with respective population-based data.

*Cascade initiated by RSV infection.* In Fig A in S1 Appendix, based on Western blotting and ELISA, we quantified levels of proteins in the cascade initiated by RSV infection at three MOIs (0.01, 0.1 and 1); in Fig B in S1 Appendix, for MOI = 0.1, we analyzed how this cascade is affected by IRF3 knockout. In brief: RSV infection leads to phosphorylation of IRF3 (Fig A panel a in S1 Appendix), which colocalizes with NF-κB in cell nuclei (Fig G in S1 Appendix), triggering synthesis and secretion of IFNβ (Fig A panel c in S1 Appendix). There are several components that form the enhanceosome and are jointly required to initiate *Ifnb1* expression, namely IRF3, NF-κB, and ATF-2/c-Jun complex [41,61]. As a simplification, in our model we included only IRF3, which localizes to cell nuclei with kinetics comparable to that of NF-κB (as we show in Fig G in S1 Appendix). The two transcription factors are well colocalized at 24 h p.i. Secreted IFNβ activates STAT1 and STAT2 (jointly represented by the accumulation of pSTATs in the model), which results in the accumulation of three proteins coded by interferon-stimulated genes: RIG-I, PKR, and OAS1 (collectively represented as ISGs in the model), as well as STAT1 and STAT2. In IRF3-deficient cells, we observed no IFNβ production (Fig A panel c in S1 Appendix) and, consequently, no STAT1/2 phosphorylation and no accumulation of ISGs.

*Dynamics of STATs and ISGs.* The IFNβ signaling is key to building an antiviral state in bystander cells, and thus we characterized in detail responses to IFNβ in separate experiments. Parameters governing pSTATs kinetics were estimated based on Fig C panels a and c in S1 Appendix. In Fig C panel a in S1 Appendix, we found that in a broad range of IFNβ stimulation doses (30 U/ml–1000 U/ml) STAT1 phosphorylation peaks and STAT2 phosphorylation plateaus 0.5 hour after stimulation. STAT2 phosphorylation monotonically increases with stimulation dose, reaching half of the maximal activation for about 100 U/ml, and this value determines the Michaelis–Menten constant in pSTATs heterodimers accumulation. Fig C panel c in S1 Appendix shows that one hour after IFNβ washout STAT1 and STAT2 lose phosphorylation, but can be rapidly re-phosphorylated in response to the next IFNβ pulse (which allows estimation of forward and reverse kinetic rate constants for pSTATs accumulation).

The forward and reverse kinetic rate constants for ISGs were estimated based on Fig C panel e in S1 Appendix, in which we observe accumulation of RIG-I, PKR, and OAS1 over 24 hours of IFNβ stimulation and their slow degradation over 48 hours after IFNβ washout and STATs dephosphorylation.

Comparison of experimental p-STAT1 and p-STAT2 time profiles with pSTATs following from the model (Fig C panels b and d in S1 Appendix) indicates that the kinetic of p-STAT1 is more complex, but the assumed simplified kinetic of pSTATs which follows the kinetic of p-STAT2 is sufficient to reproduce the kinetics of ISGs (Fig C panel f in S1 Appendix).

*Antiviral effect of IFN signaling.* In Fig D panel a in S1 Appendix we studied the importance of IFNβ and IFNλ for attenuating virus spread by knockouts of their respective receptors, IFNAR1 and IFNLR1. We observed that at 48 h p.i. there is significantly more RSV protein F in IFNAR1 KO cells than in WT cells. In turn, IFNLR1 knockout does not influence RSV protein accumulation, as indicated by a comparison of WT and IFNLR1 KO cells as well as of IFNAR1 KO cells and DKO cells (knockout of both IFNAR1 and IFNLR1). This indicates that IFNβ is critical for attenuating RSV spread, and accordingly in our RSV model the 'generic interferon' can be identified with IFNβ.

IFNβ attenuates RSV spread by activating STAT1 and STAT2, which dimerize and trigger the expression of ISGs. Accordingly, both in STAT1 KO and STAT2 KO cells we did not observe accumulation of ISGs: RIG-I, PKR, and OAS1, and observed increased accumulation of RSV protein F (Fig D panel c in S1 Appendix). Stimulation of WT cells for 24 hours before

infection with IFNβ results in a decrease of RSV protein F with respect to non-pre-stimulated cells, but this effect is negligible in STAT1 KO and STAT2 KO cells.

The effect of STAT1, STAT2, and interferon receptors knockouts on RSV propagation is reproduced by the model, as shown in Fig D panels b and d in S1 Appendix. When comparing the model with experimental data from Fig D panels a and c in S1 Appendix, we simulated IFNAR1 KO, DKO (with IFNAR1 and IFNLR1 knockouts), STAT1 KO and STAT2 KO cell by prohibiting ISGs from increasing (as in this case the p-STAT1/p-STAT2 dimers cannot be formed), while IFNLR1 KO is treated as WT (i.e., simulated with default parameters). Trajectories of p-STAT1 in STAT2 KO and p-STAT2 in STAT1 KO cells are assumed to be the same as in WT cells (as activation of STATs isoforms is independent), but, obviously, we assumed no increase of p-STAT1 in STAT1 KO cells and no increase of p-STAT2 in STAT2 KO cells.

The inhibitory effect of IFNβ prestimulation on RSV propagation is shown using Western blotting (Fig E panel a in S1 Appendix) and dPCR (Fig E panel c in S1 Appendix). Generally, the influence of prestimulation is more pronounced for lower MOIs (where the level of RSV RNA and proteins depends on replication cycles inhibited by ISGs accumulated due to IFNβ prestimulation). This tendency is reproduced by our model (Fig E panels b and c in S1 Appendix).

**Model implementation.** Agents (cells) are arranged on a triangular lattice (in which each node has 6 neighbors), which mimics a monolayer of epithelial cells (in which, based on Voronoi tessellation with geometric centers of cell nuclei used as seeds, each cell has on average 6 neighboring cells). Periodic boundary conditions are assumed along both axes. In all simulations, 100% confluency is assumed, meaning that all lattice nodes are occupied by cells.

For three variables: vRNA, pIRF3, and IFNi the last state, 3, stands for active species, while the intermediate states {1, 2} are introduced to account for a time delay associated with activation or synthesis. For the remaining three intracellular species: vProteins, pSTATs, ISGs, the intermediate states {1, 2} stand for partial activation (which reflects the assumption of gradual accumulation or activation of these species). Of note, vProteins = 1, 2, 3 may inhibit the increase of IRF3 and IFNi, but only vProteins = 3 renders the cell infectious.

The stochastic internal state of cells is coupled to an external, two-layer interferon field (see Fig 1B). The field evolves following the diffusion equation, solved using discretization introduced by cell-associated subvolumes. Cells with IFNi = 3 secrete interferon to the lower layer. Activation of pSTATs follows the Michaelis–Menten kinetics, based on the amount of interferon in this layer. The upper layer accounts for the fact that in experimental settings interferon diffusion takes place in 3D (not 2D) space.

Unless specified otherwise, the model is initialized with all stages and interferon layers set to 0. Infected cells are selected randomly with probability dictated by MOI and their v is set to 1. Prestimulation with IFN sets the upper layer to the specified concentration; a wash sets both layers to 0. A typical simulation ($100 \times 100$ grid, 100% cell confluence, IFN field update every 0.1 min) with default parameters and MOI 0.1 for 24 hours of simulated time takes a few seconds of CPU time on a standard PC.

The model does not account for cell movement, and only infection of neighboring cells is allowed. This is consistent with microscopy images where we observe growing clusters of infected cells (see Fig 2). The model also does not account for apoptosis of infected cells, as it is known that apoptosis is blocked by RSV [62]. In our experiments we see occasional formation of syncytia (see Fig 2A), which allows for a direct cell-to-cell transfer of both viral RNA and proteins. Instead of modeling syncytia formation, we have tuned our model to reflect the observed rate of infecting neighboring cells.

To report population-averaged variables, 1000 stochastic simulations were performed on the $100 \times 100$ grid, with the exception of Figs 3D and 7, and Figs A–E, Fig I, and Fig J panel a

in S1 Appendix, for which 100 stochastic simulations were performed. The estimations of credible intervals (Fig 3C) were based on 10,000 stochastic simulations.

**Comparison of experimental and simulation results.** MOI and the time post infection are two parameters that in principle should determine the state of infection. Because of unavoidable differences between experimental replicates in the number of active virus particles actually infecting cells (despite the same MOI), we observe that using the observed fraction of cells expressing RSV proteins in a given time post infection (instead of MOI) allows for more reliable comparisons across experiments, and with model simulations.

Neighboring cells are well defined for simulations (on the model lattice each cell has 6 neighbors). In order to compare simulation results with experimental data, we define neighbors by constructing Voronoi diagrams for microscope images. We identify cell nuclei using DAPI staining and use their centers of mass as Voronoi centers. We then consider cells to be neighbors if their respective Voronoi tiles share an edge. Conveniently, this procedure assigns on average 6 neighbors to each cell, the same as in our model.

In immunostaining images, levels of observed species IFNβ, nuclear IRF3, and RSV proteins are discretized into binary variables (quantification and discretization of immunostaining images are described in the previous subsection Image quantification). We perform a similar binarization in the model when comparing with immunostaining images. In the case of `IFNi` and `pIRF3`, for which the states $\{1, 2\}$ are introduced solely to account for production delay, only their last state, `3` is considered active. In the case of `vProteins` intermediate states denote the accumulation of viral proteins, and thus all states `vProteins > 0` are considered active. In the analysis of experimental data, we leave p-STAT1 as a continuous variable, while in the model we use a 4-state discretization. Importantly, this means that for low levels of p-STAT1, small differences might be picked up in experimental data but are not accounted for in the model.

To compare Western blots with the model simulation data, for a given simulation protocol we performed 100 simulations on a $100 \times 100$ lattice, and then average over all cells and simulations. We assumed that the levels of p-STAT1 and p-STAT2 are proportional to the average activation level for `pSTATs`, the levels of RIG-I, PKR, and OAS1 are proportional to the average activation level of `ISGs`, and the level of RSV protein F is proportional to the average activation level of `vProteins`. We also assumed that the level of p-IRF3 is proportional to the fraction of cells in which `pIRF3 = 3` (as in this case states `1` and `2` are introduced solely to account for time delay associated with IRF3 activation). Western blots were first normalized by the corresponding reference GAPDH, then by the maximum value in the series of measurements (from the same blot). Next, we added 0.03 to each value and finally normalized the series by the geometric mean. This procedure reflects the assumption that blots adequately capture fold differences between measurements and that differences between values smaller than 0.03 of the maximum from a given blot are not meaningful. The same normalization procedure (without the reference GAPDH step) is applied to simulation data. Overall, this approach yields a dynamic range that accommodates up to ~30-fold protein level differences.

ELISA measurements (Fig A panel c in S1 Appendix) are compared with `IFNe` summed over upper and lower compartments and averaged over cells. dPCR measurements for RSV RNA (Fig E panel c in S1 Appendix) are compared with the model based on the simplifying assumption that the average level of viral RNA is proportional to the fraction of cells with `vRNA = 3` (for `vRNA` states `1` and `2` account solely for time delay).

## Supporting information

**S1 Appendix. The Appendix PDF file containing Figs A–J and Table A.**
(PDF)

**S1 Dataset. ELISA data for Fig A panel c in S1 Appendix.**
(XLSX)

**S2 Dataset. Digital PCR data for Fig E panel c in S1 Appendix.**
(XLSX)

**S3 Dataset. Uncropped Western blot images for Fig A panel a, Fig B panel a, Fig C panel e, Fig D panel c, Fig E panel a in S1 Appendix.**
(ZIP)

## Author Contributions

**Conceptualization:** Frederic Grabowski, Marek Kochańczyk, Tomasz Lipniacki.

**Data curation:** Frederic Grabowski, Marek Kochańczyk.

**Formal analysis:** Frederic Grabowski.

**Funding acquisition:** Tomasz Lipniacki.

**Investigation:** Zbigniew Korwek, Maciej Czerkies.

**Methodology:** Zbigniew Korwek, Wiktor Prus.

**Software:** Frederic Grabowski, Marek Kochańczyk.

**Writing – original draft:** Frederic Grabowski, Marek Kochańczyk, Tomasz Lipniacki.

**Writing – review & editing:** Frederic Grabowski, Marek Kochańczyk, Zbigniew Korwek, Maciej Czerkies, Wiktor Prus, Tomasz Lipniacki.

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
