## [Decision Letter · Decision Letter 0]

23 May 2023

Dear Prof. Lipniacki,

Thank you very much for submitting your manuscript "Antagonism between viral infection and innate immunity at the single-cell level" for consideration at PLOS Pathogens. As with all papers reviewed by the journal, your manuscript was reviewed by members of the editorial board and by several independent reviewers. In light of the reviews (below this email), we would like to invite the resubmission of a significantly-revised version that takes into account the reviewers' comments.

Both reviewers found the study to be novel and of interest. However, both also raised a number of concerns. Addressing these will require changes to the text, additional model testing and, perhaps, additional experimental data.

We cannot make any decision about publication until we have seen the revised manuscript and your response to the reviewers' comments. Your revised manuscript is also likely to be sent to reviewers for further evaluation.

Sincerely,

Christopher F. Basler

Academic Editor

PLOS Pathogens

Benhur Lee

Section Editor

PLOS Pathogens

Kasturi Haldar

Editor-in-Chief

PLOS Pathogens

orcid.org/0000-0001-5065-158X

Michael Malim

Editor-in-Chief

PLOS Pathogens

orcid.org/0000-0002-7699-2064

Both reviewers found the study to be novel and of interest. However, both also raised a number of concerns. Addressing these will require changes to the text, additional model testing and, perhaps, additional experimental data.

Reviewer's Responses to Questions

**Part I - Summary**

Reviewer #1: This is a very interesting study, collecting single cell observations and using them to model RSV-induced interferon activity. The model is very interesting and distinct from models in this space. And the work suggests a new possible mechanisms of RSV inhibition of host signaling. The biggest weakness, repeated below, is that the findings are not integrated into what is known about IFN signaling. Better integration of these findings with the current literature is key.

Reviewer #2: In their manuscript, “Antagonism between viral infection and innate immunity at the single-cell level,” Grabowski et al. use high resolution in vitro data of respiratory syncytial virus (RSV) infection of A549 cells to construct a stochastic model of virus infection dynamics and the consequent cellular innate type I/III interferon (IFN) response. The model describes the reciprocal relationships of viral antagonism of the cellular IFN response, and cellular antagonism of viral replication. Among several informative vignettes, the model predicts that in RSV infection in vitro, a relatively small number of infected cells produce IFN, although IFN-producing cells are considerably enriched for infected cells.

The manuscript is extremely well-written and does an excellent job in communicating relatively complex modeling topics and interpretations such that they would be accessible to a general virology audience without sacrificing important details. The study addresses important problems regarding the dynamics of host-virus interactions and/or antagonism at single cell resolution, and importantly makes good use of empirical data to construct an informative mathematical model. The model is generally convincing, with plausible descriptions of the reciprocal antagonism between RSV and target cells. However, while informative and effective, it is not clear how much novel insight the model uncovers regarding RSV infection dynamics. Moreover, there is limited exploration and/or discussion of alternative mechanistic explanations for the experimental data and/or its incorporation into relationships modeled. Altogether, specific issues with the study are generally minor (listed below), and many can likely be addressed with additional explanation, discussion, and/or model testing:

**Part II – Major Issues: Key Experiments Required for Acceptance**

Reviewer #1: (major) please clarify for the reader how much of the model was tuned or trained using the data discussed later in the work.

(major) what was the point of the BFA treatment? As this paper will be read by modelers, some detail on what the BFA experiment reveals is important.

(major) fig 2c is not discussed in the relevant section.

(minor) the sKs metric is odd and not well know. Should provide some explanation or just use a median or mean value and report the KS p value.

(major) fig 3c, the fit to the data is pretty poor. Given the complexity of the system, this may be reasonable but worth discussing briefly in the results section.

(minor) page 6 “show that approximately 40% of RSV proteins-expressing cells…” was this a pure prediction from the model or was the model trained to try to match this data?

(major) page 7 top paragraph states that vProteins inhibitions of pIRF3 is the strongest inhibitory mechanism, implying that IFN plays a minor roll. This is concerning as several studies show that without IFN, RSV viral loads and those of other viruses are significantly higher.

(maybe important?) fig 5 A…is there any concern that the IFNbeta concentrations are clustered only in one area of the sample while RSV and IRF3 is more evenly distributed? Is this an artifact?

(major) the goodness of fit in the figures, such as 5b/c, is debatable. There do not seem to be clear trends in the data that the model replicates.

(major) the discussion needs to do a better job integrating the model’s suggestions with known experimental observations. Studies have been performed with several cell lines wherein IFN or IRF3 have been knocked out or inhibited. How do those observations compare to the findings here? As currently written, the authors are minimizing their own work by not demonstrating its importance to the field.

Reviewer #2: (No Response)

**Part III – Minor Issues: Editorial and Data Presentation Modifications**

Reviewer #1: (No Response)

Reviewer #2: -Model components.

The model includes sensible components that would be anticipated (and validated) to impact virus-cellular dynamics, including viral RNA accumulation, viral protein accumulation, phosphorylation of IRF3 (pIRF3), IFN production, and phosphorylation of STAT1/2 (pSTATs). However, viral sensing and downstream signaling can induce additional intra-/inter- cellular signaling cascades beyond IFN that can impact viral replication and cell state. While the model is necessarily (and appropriately) reductionist, have the authors considered additional, non-IFN signaling pathways that could be induced by viral infection (e.g. NfKB signaling, IL1 production, etc.)?

In addition, pIRF3 is known to induce directly the expression of several antiviral effector genes (i.e. other than IFNs) that can restrict viral functions in the infected/sensing cell. However, the model does not include a virus-inhibitory interaction for the pIRF3 term. This should be tested and/or justified.

-STAT effects on viral replication.

Modeling suggests that STAT signaling has a minimum effect of RSV propagation in this system. The authors note the “The lack of influence of the STAT signaling inhibition by RSV (nonstructural) proteins on RSV replication observed in the model suggests that this interaction may be implicated in regulatory processes not included in the model.” Given that RSV has evolved mechanisms by which to antagonize STAT signaling and consequent ISG expression (including recently described nuclear NS1 association with ISG and other gene regulatory regions, Pei et al, Cell Reports, 2021), this seems somewhat surprising, and warrants additional discussion.

-Syncytia formation

Consistent with experimental data and well-established mechanisms of the RSV life-cycle, the model only allows for infection of neighboring cells. However, in addition to “standard” cell-to-cell transmission, RSV can form eponymous syncytia, which can include the “merging” of infected cells with uninfected cells. For purposes of modeling viral spread, syncytia formation likely represents a distinct mode of transmission (e.g. potentially with considerable levels of “preformed” viral RNA and proteins “transferred” to the newly infected cell). Did the authors observe syncytia formation in the A549 infections used to support the model? In addition, how might the model be affected by/take into account syncytia formation?

PLOS authors have the option to publish the peer review history of their article (what does this mean?). If published, this will include your full peer review and any attached files.

Reviewer #1: No

Reviewer #2: No
---

## [Decision Letter · Decision Letter 1]

2 Aug 2023

Dear Prof. Lipniacki,

We are pleased to inform you that your manuscript 'Antagonism between viral infection and innate immunity at the single-cell level' has been provisionally accepted for publication in PLOS Pathogens.

Best regards,

Christopher F. Basler

Academic Editor

PLOS Pathogens

Benhur Lee

Section Editor

PLOS Pathogens

Kasturi Haldar

Editor-in-Chief

PLOS Pathogens

orcid.org/0000-0001-5065-158X

Michael Malim

Editor-in-Chief

PLOS Pathogens

orcid.org/0000-0002-7699-2064

We recommend having a colleague proofread the manuscript to address any grammatical issues. Some suggested edits are as follows (line numbering based on the tracked changes version of the revised ms):

Line 182 change disjoint to “disjointed”. Would “discordant” be a better word?

Line 191 change “this is not our case.” To “this is not the case in our study.”

Line 258 edit to “As previously demonstrated in…

Lines 261-265. Consider rewording this section to improve its clarity.

Line 272 Change “producing jointly” to “jointly producing”

Line 360 Change "Consistently" to "Consistent"

Reviewer Comments (if any, and for reference):

Reviewer's Responses to Questions

**Part I - Summary**

Reviewer #1: The author's nicely addressed my concerns.

**Part II – Major Issues: Key Experiments Required for Acceptance**

Reviewer #1: None

**Part III – Minor Issues: Editorial and Data Presentation Modifications**

Reviewer #1: None

PLOS authors have the option to publish the peer review history of their article (what does this mean?). If published, this will include your full peer review and any attached files.

Reviewer #1: No

---

## [Editor Report · Acceptance letter]

30 Aug 2023

Dear Prof. Lipniacki,

We are delighted to inform you that your manuscript, "Antagonism between viral infection and innate immunity at the single-cell level," has been formally accepted for publication in PLOS Pathogens.

Best regards,

Kasturi Haldar

Editor-in-Chief

PLOS Pathogens

orcid.org/0000-0001-5065-158X

Michael Malim

Editor-in-Chief

PLOS Pathogens

orcid.org/0000-0002-7699-2064